# Low Carbon Energy Generates Public Health Savings in California

Christina B. Zapata[1], Chris Yang[2], Sonia Yeh[2], Joan Ogden[2], Michael J. Kleeman[1]

[1] Department of Civil and Environmental Engineering, University of California – Davis, Davis, California, USA
[2] Institute of Transportation Studies, University of California – Davis, Davis, California, USA

Correspondence to: Michael J. Kleeman (mjkleeman@ucdavis.edu)

**Abstract.** California's goal to reduce greenhouse gas (GHG) emissions 80% below 1990 levels by the year 2050 will require adoption of low carbon energy sources across all economic sectors. In addition to reducing GHG emissions, shifting to fuels with lower carbon intensity will change concentrations of short-lived conventional air pollutants, including airborne particles with diameter less than 2.5 $\mu$m ($PM_{2.5}$) and ozone ($O_3$). Here we evaluate how business-as-usual (BAU) air pollution and public health in California will be transformed in the year 2050 through the adoption of low-carbon technologies, expanded electrification, and modified activity patterns within a low carbon energy scenario (GHG-Step). Both the BAU and GHG-Step state-wide emission scenarios were constructed using the energy-economic optimization model, CA-TIMES, that calculates the multi-sector energy portfolio that meets projected energy supply and demand at the lowest cost, while also satisfying scenario-specific GHG emissions constraints. Corresponding criteria pollutant emissions for each scenario were then spatially allocated at 4 km resolution to support air quality analysis in different regions of the state. Meteorological inputs for the year 2054 were generated under a Representative Concentration Pathway (RCP) 8.5 future climate. Annual-average $PM_{2.5}$ and $O_3$ concentrations were predicted using the modified emissions and meteorology inputs with a regional chemical transport model. In the final phase of the analysis, mortality (total deaths) and mortality rate (deaths per 100,000) were calculated using established exposure-response relationships from air pollution epidemiology combined with simulated annual-average $PM_{2.5}$ and $O_3$ exposure. Net emissions reductions across all sectors are -36% for $PM_{0.1}$ mass, -3.6% for $PM_{2.5}$ mass, -10.6% for $PM_{2.5}$ EC, -13.3% for $PM_{2.5}$ OC, -13.7% for NOx, and -27.5% for $NH_3$. Predicted deaths associated with air pollution in 2050 dropped by 24%–26% in California (1,537–2,758 avoided deaths yr$^{-1}$) in the "climate-friendly" 2050 GHG-Step scenario, which is equivalent to a 54%–56% reduction in the air pollution mortality rate (deaths per 100,000) relative to 2010 levels. These avoided deaths have an estimated value of $11.4B–$20.4B USD per yr$^{-1}$ based on the present-day Value of a Statistical Life (VSL) equal to $7.6M. The costs for reducing California GHG emissions 80% below 1990 levels by the year 2050 depend strongly on numerous external factors such as the global price of oil. Best estimates suggest that meeting an intermediate target (40% reduction in GHG emissions by the year 2030) using a non-optimized scenario would reduce personal income by $4.95B yr$^{-1}$ (-0.15%) and lower overall state GDP by $16.1B yr$^{-1}$ (-0.45%). The public health benefits described here are comparable to these cost estimates, making a compelling argument for the adoption of low carbon energy in California, with implications for other regions in the United States and across the world.

## 1 Introduction

Implementation of California's climate policy (Executive Order S-3-05) to reduce GHG emissions 80% below 1990 levels by the year 2050 will require widespread adoption of low-carbon energy supply and demand technologies across the state's entire economy. These changes will not only reduce California's contribution to climate change, they will also alter the chemical composition, spatial pattern, and attributable adverse health effects of the state's serious air pollution problem. Reducing long-term exposure to fine airborne particulate matter ($PM_{2.5}$) and ozone ($O_3$) will improve public health through a reduction in premature mortality (Krewski, Jerrett et al. 2009, Lepeule, Laden et al. 2012).

California's near-term measures to mitigate greenhouse gas (GHG) emissions are required by the Global Warming Solutions Act of 2006 (Assembly Bill (AB) 32). Since the adoption of AB 32, a wave of incentives, mandates, carbon markets, fees, and standards have been implemented to curb the rate of the state's GHG emissions. Regulations include the Renewable Portfolio Standard (RPS) for the electricity generation sector, the Low Carbon Fuel Standard (LCFS) aimed at reducing carbon intensity of transport fuels, the Pavley Clean Car Standards for fuel economy and $CO_2$ emissions, and the Cap-and-Trade Program. Zapata et al. (2012) analysed the air quality co-benefits of AB 32 and found that the GHG mitigation measures had the co-benefit of reducing $PM_{2.5}$ concentrations in California by ~6% in the year 2030 with a corresponding decrease in air-pollution mortality. Additional measures will be needed to meet the targets included in California's Executive Order S-3-05 that calls for GHG emissions to decrease 80% below 1990 GHG levels by the year 2050.

Numerous previous studies have examined the relationship between climate policies and air quality using methods tailored to match the region of interest (Table S1). For example, Jacobson et al. (Jacobson, Delucchi et al. 2014, Jacobson, Delucchi et al. 2015) examined how a scenario of 100% wind, water, and solar would alter all economic sectors leading to changes in air quality and health impacts for California and the United States in 2050. This bounding analysis is extremely valuable since it quantifies the maximum possible air quality benefits associated with climate policies but a recent analysis suggests that scenarios incorporating a broader range of technologies are may be more realistic (Clack, Qvist et al. 2017). The debate on this point is ongoing (Jacobson, Delucchi et al. 2017). For studies that consider a broad range of technologies, mMultiple approaches have been used to select between the diverse technologies available in these future scenarios, but the majority of these studies rely on the expert opinions of the authors rather than an objective analysis. For example, Shindell, Kuylenstierna et al. (2012) created a future scenario by selecting measures that were "assumed to improve air quality" and mitigate both long-lived GHGs and short-lived criteria pollutants after ranking them by climate impact. The extensive study by van Aardenne, Dentener et al. (2010) explored 6 scenarios with wider levels of air and/or climate policy, as well as the option of biofuel consumption, however, technology adoption is again largely dependent on author specified assumptions on shares of existing technologies. Since the technology choices in each scenario strongly affect the air quality outcomes, the author assumptions in these previous studies have a strong influence on the calculated health benefits stemming from reduced air pollution concentrations. As a secondary limitation, many previous studies have been carried out for regions much larger than California which requires the use of coarse grid cells that do not completely resolve

important spatial patterns of pollutants within the state's complex topography (West, Smith et al. 2013, Garcia-
Menendez, Saari et al. 2015).
Here we build on the previous work on climate policy – air quality interactions by conducting an optimized
emissions analysis at high spatial resolution for California.  The state of California has a very large and diverse
economy and so it is difficult to design optimal GHG mitigation strategies using expert opinions alone.  Energy -
economic optimization models are needed to find least-cost scenarios that achieve GHG objectives within the
resource constraints of the state.  California also has significant existing environmental regulations and so detailed
analysis is required to account for the impact of technology, fuel, and behavioural changes implied by broad GHG
policies on the landscape of pre-existing rules.  All of this analysis must be carried out at high spatial resolution to
properly calculate air pollution exposure in major cities that often experience a sharp gradient of pollutant
concentrations across their boundaries.
Zapata et al. (Zapata, Yang et al. 2017) used the CA-REMARQUE (CAlifornia REgional Multisector AiR QUality
Emissions) model to predict criteria pollutant emissions associated with two economically optimized scenarios for
California in the year 2050: (i) a Business-as-Usual (BAU) scenario that includes all existing environmental laws in
California including AB 32 and (ii) a greenhouse gas mitigation (GHG-Step) scenario including additional least-cost
policy and technology adoption needed to achieve the 80% GHG reduction objective of Executive Order S-3-05
using a $CO_2$ constrained step function. The results indicated that adoption of the measures in the GHG-Step scenario
could cause decreases or increases in criteria pollutant emissions in different economic sectors/locations due to the
trade-offs involved in the state-wide cost minimization approach.  As a further complication, switching to alternative
lower carbon intensive fuels in the GHG-Step scenario altered the composition of reactive organic gas (ROG)
emissions and the size and composition of particulate matter emissions.  These finding re-enforce the need for
sophisticated analysis methods in complex regions like California.
The overall goal of the present study is to quantify air pollution and health implications associated with the BAU and
GHG-Step scenarios described by Zapata et al. (Zapata, Yang et al. 2017) acting across the entire California energy-
economy in the year 2050. The air pollution concentrations associated with the BAU and GHG-Step scenarios are
calculated at 4 km resolution using a regional chemical transport model and the avoided mortality is estimated using
established relationships from air pollution epidemiology.  Economic benefits are then calculated with the Value of a
Statistical Life (VSL).  Finally, the total public health benefits from avoided air pollution are compared to the total
incremental cost for adoption of low carbon energy in California to better understand the net costs for the GHG
mitigation program.

## 2 Methodology

Air quality and health impacts associated with energy scenarios in the year 2050 were determined by combining
estimated changes to criteria pollutant emissions inventories with downscaled meteorology as inputs to a regional air
quality model to predict air quality with 4 km resolution over California.  Epidemiology risk exposure functions and
mortality data were then used to estimate premature deaths. Figure 1 summarizes the calculations with additional
details provided below.

### 2.1 Criteria Pollutant Emissions

Criteria pollutant emissions were predicted with the California REgional Multisector AiR QUality Emissions (CA-
REMARQUE) model (Zapata, Yang et al. 2017) for the BAU and GHG-Step scenarios. Both scenarios were
constructed using CA-TIMES, a technology-rich, bottom-up, energy economics model that determines the least-cost
mix of technology/fuel options for all sectors of the state-wide economy. CA-REMARQUE translated these
behaviour, technology, and fuel changes into spatially- and temporally-resolved criteria pollutant emissions
inventories. CA-REMARQUE predicted that adoption of the GHG-Step policies in place of the BAU policies
would cause decreases in emissions of primary $PM_{0.1}$ (-36%), $PM_{2.5}$ (-43.6%), oxides of nitrogen ($NO_x$, -13.74%),
and ammonia ($NH_3$, -27.58%) but cause increases in emissions of carbon monoxide (CO, +37%) and oxides of
sulfur (SOx, +14%). Some components of primary $PM_{2.5}$ emissions responded more strongly to different
technology changes yielding non-uniform reductions of $PM_{2.5}$ elemental carbon (EC, -10.64%), $PM_{2.5}$ organic
carbon (OC, -13.3%), and $PM_{2.5}$ copper (Cu, -63%). The spatial allocation of emission rates was determined by
either using existing 4km spatial patterns of emissions sources or finding new optimal locations for new emissions
sources such as biorefineries that were placed near high biomass feedstock regions. The future BAU and GHG-Step
scenarios considered in the present study do not include nuclear or coal-fired (with or without CCS) electricity
generation in California. Electricity generation in the 2050 GHG-Step scenario is dominated by wind (34%), solar
(34%), and natural gas (18%) with smaller contributions from tidal, geothermal, and hydro. A comprehensive
analysis of all emissions changes including spatial plots is provided by Zapata et al. (Zapata, Yang et al. 2017).

### 2.3 Meteorology Fields

Meteorology simulations using the Weather Research and Forecasting (WRF) model v3.2.1 (University Corporation
of Atmospheric Research 2010) conducted previously (Zhang, Chen et al. 2014) for years 2048–2054 were used as
meteorological inputs in this study. Hourly-averaged fields describing spatial and temporal wind speed and direction,
humidity, temperature, planetary boundary layer (PBL) height, downward shortwave radiation, air density, and
precipitation were formatted for use with the regional chemical transport model. The 2054 calendar year was the
median year over the period 2048–2054 for domain-average PM2.5 concentrations within the South Coast Air Basin
that contains the majority of the population in California. The 2054 calendar year was selected as the median year
within the 2048–2054 timespan based on population-weighted average $PM_{2.5}$ exposure with the least deviation from
the 7-year episodic mean (Mahmud, Hixson et al. 2010) based on analysis using the 2010 CARB emission inventory
(Zhang, Chen et al. 2014). The focus of the current study is to evaluate how the emissions changes lead to different
air quality outcomes. Both emissions scenarios are evaluated using the same meteorology, which minimizes the
variability introduced by the climate signal.

### 2.4 Regional Chemical Transport Model Configuration and Simulation

Air quality was simulated using the UCD-CIT (University of California, Davis – California Institute of Technology) 3D regional chemical transport model (Kleeman and Cass 2001, Ying, Fraser et al. 2007, Hu, Zhang et al. 2015)). The SAPRC11 (Carter and Heo 2012, Carter, Heo et al. 2012) chemical mechanism was used to represent gas-phase chemical reactions. Gas-to-particle conversion was simulated as a dynamic process based on the concentration of semi-volatile gas-phase compounds at the particle surface in equilibrium with the condensed material inside each particle. Thermodynamic equilibrium within each particle for inorganic species was calculated using the ISORROPIA model (Nenes et al, 1998). Thermodynamic equilibrium within each particle for organic species was calculated using a two-product model (Carlton et al., 2010). PM emissions profiles include 18 organic, inorganic, and metal particulate species distributed across 15 size bins.

Air quality simulations were conducted over three horizontal domains, a coarse 24 km parent domain, and two 4 km resolution child domains. The coarse domain covered all of California and the adjacent Pacific Ocean to provide boundary inputs to the higher resolution child domains over populated regions in northern and southern California. Sixteen telescoping vertical layers were used up to a total height of 5km above ground. Simulations were conducted for the first 28–29 days of each month for the 2054 calendar year. The first 3 days of every month were excluded to minimize the effects of initial conditions which are not known exactly, leaving 301 of simulation days to be used in the statistical analysis.

### 2.5 Population Projections

A 2050 California population projection at 4 km spatial resolution was used for both population-weighted concentration estimates and mortality estimates. This population projection is based on the highly-resolved block-group 2010 Census population data in shapefile format (U.S. Census Bureau) which was intersected with the regular air quality grid. The 4 km resolution population field was then scaled according the projected populations for each county in 2050 (California Department of Finance. Demographic Research Unit 2014) relative to 2010 (Table S2). This procedure was conducted separately for population age >35 and for all ages (see Fig. S1) to be used for the population-weighted code (all ages) and the mortality estimates (>35 years). The combined southern and northern 4 km resolution modeling domains encompassed 92% of California's projected 2050 population (summarized in Table S3).

Population acts as a spatial surrogate for distributing emissions and as a receptor for calculating the public health effects of air pollution. Consistent population fields were used for both of these tasks in the current study. Population growth rates by county are summarized by (Zapata, Yang et al. 2017).

### 2.6 Statistical and Exceedance Analysis

Several statistical analyses were conducted across space, seasons, and scenarios. Annual average concentration plots were estimated by taking the average of 301 daily concentrations fields. A two-tailed paired t-test was used to identify significant differences between BAU and GHG-Step concentrations. Annual or seasonal concentration field plots

were condensed to a state-wide, air basin, or county population-weighted concentration estimate by summing the
concentration multiplied by the population in each cell and then dividing the resulting sum by the entire population
for the region of interest.
Daily maximum 8-h average $O_3$ concentrations were calculated for each model grid cell. Subsequent seasonal or
annual averages used the daily maximum 8-h average concentrations for a given state, basin, or county. To determine
whether a county was in compliance with the 70 ppb $O_3$ National Ambient Air Quality Standards (NAAQS), the fourth
highest population-weighted maximum 8-h $O_3$ concentration was calculated. The number of days exceeding this
standard was also tabulated.
**2.7 Mortality and Cost Estimates**
Premature mortality estimates from long-term exposure to $PM_{2.5}$ and $O_3$ were calculated using annual-average 4 km
resolution concentration fields for the BAU and GHG-Step scenarios. The attributable fraction (AF) is the portion of
deaths or incidences that can be associated with the cause of interest, in this case the fraction of deaths due to annual
$PM_{2.5}$ and $O_3$ exposure. The AF quantifies the change in the relative risk.
$$AF_i = \frac{RR_i - 1}{RR_i} = \frac{e^{\beta(x_i - x_{i,bkg})} - 1}{e^{\beta(x_i - x_{i,bkg})}}$$ (1)
The log-linear incidence rate function is assumed when calculating the risk ratio (RR) as shown in Eq. (1). The beta
coefficient ($\beta$), is derived from taking the natural log of the RR found in epidemiology literature. $PM_{2.5}$ RR for all-
cause mortality associated with a 10 µg m$^{-3}$ increase in long-term $PM_{2.5}$ exposure is estimated at 1.062 based on an
worldwide meta-analysis (Hoek, Krishnan et al. 2013) or 1.036 based on the American Cancer Society follow-up
(Krewski 2009). An $O_3$ RR of 1.04 for respiratory mortality from long-term $O_3$ exposure is based on (Jerrett, Burnett
et al. 2009). The change in concentration is based on taking the annual average concentration for a given grid cell ($x_i$)
and subtracting it from the background concentration ($x_{i,bkg}$). Background concentrations on the west coast of North
America are often measured at mountain sites that sample the free troposphere. Herner et al. (Herner, Aw et al. 2005)
measured PM1.8 concentrations of 4 µg m$^{-3}$ at Sequoia National Park (elevation 535m) during periods when this site
was in the free troposphere. McKendry (McKendry 2006) surveyed published literature and reviewed monitoring
data in British Columbia on the west coast of North America and estimated that background PM2.5 concentrations
are 2 µg m$^{-3}$ with little evidence of change over time. A background $PM_{2.5}$ concentration of 3 µg m$^{-3}$ (Ostro 2004) and
$O_3$ concentration of 35 ppb was assumed in the current study. The beta coefficient, change in cell concentration, is
then used to calculate the risk ratio ($RR_i$) and subsequently the attributable fraction.
$$E_s = \sum_i AF_i \, B_c \, P_i$$ (2)
The mortality ($E_s$) for each scenario for a given region, was calculated using Eq. (2) by taking the product of the
population and mortality rate to get the deaths, followed by multiplying the fraction that is attributable to pollution
(see Eq. (1)). Population ($P_i$) projections for ages 35 and older were used in this calculation due to high uncertainty
for younger age groups. Averaged 2009–2013 California all-cause (all ICD 10 codes) and respiratory (ICD 10 codes
J0–J98) mortality rates ($B_c$), calculated in deaths per 100,000, were determined for each California county for ages 35
and older from the CDC WONDER database (United States Department of Health and Human Services (US DHHS),
Centers for Disease Control and Prevention (CDC) et al. 2014).
Costs associated from premature death from long-term air pollution exposure were estimated using the "value of a
statistical life" (VSL) method, assuming that a death equates to $7.6M USD, based on the distribution of 26 economic
reports (Viscusi and Aldy 2003) and the suggested value by the EPA (Industrial Economics 2011, Ostro 2015, RTI
International 2015). This value can be adjusted to a future year with an average discount rate "$i$" by multiplying with
the value $(1+i)^{future\ year-base\ year}$ where base year is 2006. VSL is estimated based on a "willingness to pay" for small
reductions in mortality risk through the selection of different job types. "Willingness to pay" estimates are thought to
incorporate "cost of illness" including morbidity but they do not capture non-health damage.
**3 Results and Discussion**
**3.1 Ozone (O$_3$) Concentration**
**3.1.1 Annual Average and Seasonal Ozone Changes**
Figure 2a shows the population-weighted daily maximum 8-h ozone concentrations for the 2054 meteorological year
under the BAU and GHG-Step emissions scenarios. Box and whisker plots are shown for winter, summer, and annual
time periods to consider both cyclical and yearly effects. Figure 3a illustrates the spatial distribution of ozone
concentrations in the BAU scenario while Figure 3b illustrates the changes induced by the GHG-Step scenario. The
annual-average BAU 8-h ozone concentration reaches a maximum of 61 ppb in Southern California downwind (east)
of Los Angeles and San Bernardino. In the northern/central California domain, the annual-average BAU 8-h ozone
concentration has a maximum value of 57 ppb along the Northern Central Coast air basin, around Santa Clara and San
Benito County.
Figure 3b illustrates that regional 8-h average ozone concentrations (annually averaged) in the San Joaquin Valley
(SJV) air basin (containing the cities of Bakersfield and Fresno) decrease by 2 ppb–3 ppb under the GHG-Step
scenario. GHG mitigation strategies did not reduce ozone concentrations in major population centres including the
San Francisco (SF) air basin and the South Coast (SC) air basin (containing the city of Los Angeles). To the contrary,
ozone concentrations increased in these dense urban regions because BAU conditions have excess NO$_x$ concentrations
that titrate ozone. The extent of NO$_x$ emission reductions under the GHG-Step scenario is insufficient to shift the
chemical regime to one where decreases in NO$_x$ lead to O$_3$ reductions, instead favouring more ozone formation
(Seinfeld and Pandis 2006).
Figure 2a illustrates that population-weighted annual average 8-h ozone concentrations in the rural SJV decreased by
-4.3% (52 ppb to 50 ppb) in the GHG-Step scenario with the greatest reductions occurring in the summer months (-
9.4%). In contrast, population weighted annual average 8-h ozone concentrations increased in urbanized regions (SC
+5.1%, SD +2.8%, SF +6.5%) consistent with the regional trends illustrated in Figure 3b. Population-weighted ozone
concentrations under the GHG-Step scenario increased in SC, SD, and SF during winter (+7.0 %, +9.3 %, and +17 %,
respectively) but had mixed trends during summer: ozone concentrations in SC and SF (highest population density)
increased by +3.2% and +6.1%, respectively, under the GHG-Step scenario but concentrations in SD (slightly lower
population density) decreased -2.2% during the summer season.
Overall, a state-wide increase of +3.9 % in population-weighted annual-average 8-h ozone concentrations occurred
under the GHG-Step scenario because increased ozone concentrations in heavily populated SF, SC, and SD
overwhelmed decreased ozone concentrations in the SJV. The regulatory and health implications of this finding will
be discussed in subsequent sections.
**3.1.2 High Ozone Events and Number of Exceedance Days**
Most benchmarks for ozone concentrations decrease strongly across California in the 2050 BAU scenario relative to
current 2010 levels. Simulations carried out using identical 2010 summer meteorological fields but different
emissions inputs (2010 vs. 2050) demonstrate that emission changes - rather than weather inputs - were the primary
cause of these decreasing $O_3$ concentrations. Table 1 summarizes the 4th highest maximum 8-h average ozone
concentration and the number of days exceeding the 70 ppb 8-h average ozone standard for different California
counties. The 4th highest 8-h average $O_3$ concentration of each year, averaged over 3 years is used to determine if a
given area is in compliance with the NAAQS. Many California air districts violate the 8-h $O_3$ NAAQS, with
classifications ranging from moderate, serious, severe or extreme levels of $O_3$ (Table S4). The county median of the
4th highest 8-h simulated ozone concentration in 2010 is 92.2 ppb (IQR: 74.0 ppb–99.1 ppb) with 23 out of 26
counties analyzed reaching levels ≥ 70 ppb. The county median of the 4th highest 8-h average ozone concentration
in the 2050 BAU scenario decreases to 69.2 ppb (IQR: 66.2 ppb–71.9 ppb) with a further decrease to 64.2 ppb (IQR:
62.8 ppb–66.4 ppb) in the GHG-Step scenario.
Almost half (10 of 23) of the counties exceeding the $O_3$ NAAQS in 2010, would achieve attainment with the
standards in the 2050 BAU scenario and nearly all (19 out of 23) counties would achieve attainment under the 2050
GHG-Step scenario. Only the SC counties of Los Angeles, Orange, Riverside, San Bernardino are predicted to
remain out of attainment with the ozone NAAQS in the 2050 GHG-Step scenario.
As noted above, some regions experience ozone dis-benefits under the GHG-Step scenario which has implications
for compliance with the ozone NAAQs. Table 1 illustrates that increases in the 4th highest 8-h ozone concentrations
under the GHG-Step scenario may prevent Orange and Los Angeles counties from complying with the 70 ppb
standard. The 4th highest 8-h ozone concentrations in San Bernardino County would not comply with the $O_3$
NAAQS under either emissions scenario, with concentrations increasing from 80 ppb in the BAU scenario to 82 ppb
in the GHG-Step scenario. Both San Francisco and San Mateo counties were predicted to experience higher ozone
concentrations in the GHG-Step scenario but would remain in compliance, with maximum concentrations of 63 ppb
and 61 ppb, respectively.
Figure 4 illustrates the number of days exceeding the 8-h ozone standard of 70 ppb in California under 2010
conditions, the 2050 BAU scenario, and the 2050 GHG-Step scenario. Most counties in central California have ~60
ozone exceedance days in 2010, ~5–10 ozone exceedance days in the 2050 BAU scenario, and zero ozone
exceedance days in the 2050 GHG-Step scenario. North Central Coast (NCC) basin ozone reductions at Monterey,
San Benito, and Santa Cruz counties also enabled those counties to comply with the $O_3$ standards in the GHG-Step
scenario. The relatively small increase in ozone exceedance days in southern California counties like Los Angeles,
Orange, San Bernardino, and San Diego will require extra mitigation strategies to achieve compliance with the
ozone NAAQS.
**3.2  PM2.5 Mass Concentration**
$PM_{2.5}$ concentrations can be analyzed on time scales ranging from seconds to years, but annual average $PM_{2.5}$
concentrations are most commonly used to calculate mortality and health damages. Figure 5 illustrates annual average
$PM_{2.5}$ concentrations in Northern/Central California and Southern California in 2054 under the BAU scenario (Figure
5a) and the differences induced by the GHG-Step scenario (Figure 5b). Both results use identical 2054 meteorology,
ensuring that the concentration differences reflect changes between each scenario's emissions inventory. The highest
BAU annual-average $PM_{2.5}$ concentration in southern California is ~18 µg m$^{-3}$ in the city of San Bernardino located
east of Los Angeles, with the next highest $PM_{2.5}$ hot spots occurring at San Diego, and near the busy Port of Los
Angeles/Long Beach. In Northern California, the annual-average $PM_{2.5}$ peaks at 25.3 µg m$^{-3}$ between the cities of
Oakland and San Francisco (SF). Maximum $PM_{2.5}$ reductions in the GHG-Step scenario (Figure 5b) occur between
Oakland and San Francisco (-6 µg m$^{-3}$), in San Diego county (-5.3 µg m$^{-3}$), and in San Bernardino county (-3.5 µg m$^{-3}$
). Overall, the reductions are significant (p-value $\leq$ 0.1) over the majority of Northern and Southern California; the
only non-significant $PM_{2.5}$ changes are two locations inland in northern Los Angeles around Lancaster and in
Midwestern San Bernardino where BAU concentrations were low. Significant $PM_{2.5}$ increases of +0.5 µg m$^{-3}$ do
occur in ocean shipping routes because more fossil fuel is used for marine vessels in the GHG-Step scenario than in
the BAU scenario. The GHG-Step scenario requires increased bio-fuel use as part of the overall strategy to reduce
GHG emissions. This increased biofuel production is associated with higher biofuel costs since the least expensive
biofuel feedstocks are used first followed by progressively more expensive feedstocks. As biofuels utilization
increases, the demand and cost for conventional fossil fuels decreases. The decreased cost for fossil fuels in the GHG-
Step scenario makes these fuels attractive for use by marine sources.
Population-weighted $PM_{2.5}$ concentrations (Fig. 2b) decrease for all regions in all seasons under the 2050 GHG-Step
scenario relative to the BAU scenario. Variability in $PM_{2.5}$ concentrations is highest during the winter, with periods
of intense stagnation intermixed with periods of vigorous atmospheric mixing. $PM_{2.5}$ concentrations are less variable
in the summer months as demonstrated by the smaller Inter Quartile Range (IQR) in Figure 2b. The annual population-
weighted $PM_{2.5}$ concentration drops from 6.0 to 4.8 µg m$^{-3}$ (-20%) in the San Joaquin Valley (SJV), 8.3 to 6.2 µg m$^{-3}$
(-25%) in San Diego (SD), 9.5 to 7.8 µg m$^{-3}$ (-18%) in San Francisco Bay Area (SF), and 7.6 to 6.5 µg m$^{-3}$ (-14%) for
the South Coast (SC) air basins. Additional detail of the $PM_{2.5}$ species that decreases the most (e.g. nitrate) and the
changes in the particulate size distribution are further described in SI and summarized in Table S5.
Certain $PM_{2.5}$ spatial patterns illustrated in Figure 5 were difficult to anticipate based exclusively on state-wide
emissions totals. For example, the $PM_{2.5}$ co-benefits from wide-spread adoption of new vehicle technology contribute
significantly to state-wide emissions reductions, but these changes were distributed over a larger area than the benefits
associated with the decarbonization of freight modes (e.g. rail, aviation and marine). Most on-road vehicles in
California already have relatively low emissions rates for criteria pollutants. Further vehicular emissions savings
result from small reductions that are distributed over the large number of vehicles across the entire state. This spreads
the air quality improvements associated with vehicles over a large area. In contrast, freight modes use fuel with higher
sulfur content burned in engines with less aftertreatment control (e.g. particulate filter) leading to higher PM emission
rates per energy consumed (e.g. mg J$^{-1}$). These sources are localized to goods movement corridors (shipping lanes,
rail lines, etc.) that intersect at transport distribution hubs near ports. This leads to localized reductions in particulate
matter concentrations associated with freight modes compared to more diffuse reductions associated with on-road
sources. These trends were not obvious from state-wide emissions tables but are clearly illustrated by the results from
regional air quality modeling.
**3.3 Associated PM$_{2.5}$ and O$_3$ Mortality, Mortality Rate, and Costs**
Figures 6 and 7 illustrate the deaths, death rate, and cost associated with premature deaths from long-term annual
exposure of both PM$_{2.5}$ and ozone (O$_3$).
**3.3.1 Mortality**
County and state-wide PM$_{2.5}$ and O$_3$ associated deaths are displayed in Figure 6a and Figure 7a. The calculations
summarized in Figure 7a predict that 6,400–10,600 people would die annually in the California 2050 BAU scenario
due to exposure to PM$_{2.5}$ and O$_3$ depending. The medium estimate for mortality falls between these low and high
estimates. The rangeis estimate includes population growth through 2050. In the California GHG-Step scenario, total
PM$_{2.5}$ and O$_3$ mortality would decrease to 4,800–7,900 deaths annually (24%–26% reduction) due to reductions in
pollutant concentrations. More than 95% of the premature mortality is associated with PM$_{2.5}$ while only 2.0%–4.4%
is attributed to O$_3$. As a result, the O$_3$ increases associated with the GHG-Step scenario have a minor effect on
mortality relative to PM$_{2.5}$. Spatial trends for PM$_{2.5}$ and O$_3$ mortality are similar, with the highest rates occurring in
highly populated regions (see Figure 3a and Figure 5a). Likewise, most of the avoided mortality in the GHG-Step
scenario also occurs in the regions with the highest populations.
**3.3.2 Mortality Rate**
Air pollution mortality rates (deaths per 100,000 people) plotted in Figure 6b and Figure 7b help to compare health
effects across urban and rural areas (both of which can experience high pollution events in California). The 2050
state-wide air pollution mortality rate drops by 54%-56% in the 2050 GHG-Step scenario vs. the 2010 scenario and
24%–26% in the GHG-Step scenario vs. the BAU scenario. Reductions in the air pollution mortality rate were
predicted in all counties under the GHG-Step scenario vs. the BAU scenario (Figure 6b). In the 2050 BAU scenario,
San Francisco, San Mateo, Alameda, Contra Costa, Sacramento, San Diego and San Bernardino counties are predicted
to have air pollution mortality rates higher than the state-wide average of 19.3–32.2 deaths per 100k people (see Figure
6b). Under the GHG-Step scenario, San Francisco, San Mateo, and Alameda counties continue to have the highest
death rates associated with PM$_{2.5}$ and O$_3$. Mortality rates in SF are more than double the state-wide average due to
the proximity of major construction projects and growing populations. Overall, Sacramento, Solano, Contra Costa,
and San Francisco counties are predicted to have the greatest reduction in $PM_{2.5}$ and $O_3$ mortality rates due to the
adoption of GHG mitigation strategies. These patterns reflect a reduction in the emissions of criteria pollutants from
construction projects but an increase in emissions from locations that produce new energy sources such as biofuels.
$O_3$ mortality is expected to increase from 260 deaths $yr^{-1}$ in the BAU scenario to 490 deaths $yr^{-1}$ in the GHG-Step
scenario due to the increase of $O_3$ in key populated areas (mainly greater Los Angeles). The largest number of $O_3$
associated deaths (~25%) are estimated to occur in southern California due to the combination of high population and
excess $NO_x$ in the BAU scenario leading to increased $O_3$ concentrations when $NO_x$ emissions decrease in the GHG-
Step scenario. The portion of air pollution deaths due to $O_3$ would increase from 2.4%–4% in the BAU scenario to
6.2%–10.1% in the GHG-Step scenario, but overall mortality still decreases due to the overwhelming effect of $PM_{2.5}$
reductions.

### 3.4 Benefits

Using a Value of a Statistical Life (VSL) equal to $7.6M per avoided death (Industrial Economics 2011, Ostro 2015),
total costs for premature deaths in California equal ~$47.0B–$78.5B per year in the 2050 BAU emissions scenario,
with a savings of $11.4B–$20.4B per year in the GHG-Step emissions scenario (right axis Figure 7a). Los Angeles
County has the highest premature mortality associated with air pollution (25% of California) and thus the highest air
pollution mortality cost under all emissions scenarios. Air pollution damages in Los Angeles County are valued at
$15.2B–$25.5B per year in 2010, which decreases to $12.1B–$19.6B per year in 2050 BAU. Adoption of the GHG
mitigation strategies in California reduces air pollution damages in Los Angeles County by $1.9B–$3.6B per year
(17%–18% reduction). Other major counties also experience reduced air pollution costs under the GHG-Step scenario
relative to BAU, including San Diego ($1.7B–$2.9B per year reduction; 15%–16%), and Sacramento ($0.70B–$1.3B
per year reduction; 6.4%). However, the largest cost savings per capita are predicted to occur in and around counties
near San Francisco based on the higher mortality rate reductions.

### 3.5 Implications

The costs for reducing California GHG emissions 80% below 1990 levels by the year 2050 depend strongly on
numerous assumptions about external factors such as the global price of oil. Only a few California energy models are
available that attempt to calculate costs across the entire economy (Morrison, Eggert et al. 2014, Morrison, Yeh et al.
2015). Analysis produced by the E3 Pathways model (Williams, DeBenedictis et al. 2012, Energy+Environmental
Economics (E3) 2015) suggest that meeting an intermediate target (40% reduction in GHG emissions by the year
2030) using a non-optimized energy portfolio scenario would reduce personal income by $4.95B $yr^{-1}$ (-0.15%) and
lower overall state GDP by $16.1B $yr^{-1}$ (-0.45%). Analysis produced by the CA-TIMES model (Yang, Yeh et al.
2014, Yang, Yeh et al. 2015) indicates that the optimized GHG-Step scenario is less expensive than the BAU
scenarios.
The air pollution analysis carried out in the current study predicts that the GHG-Step scenario will provide public
health benefits equivalent to $11.4B–$20.4B per year relative to the BAU scenario in 2050. The public health benefits
described here have relatively tight uncertainty ranges with median values that are comparable to the more pessimistic
of these two cost estimates for the adoption of low carbon energy.
Figure 8 illustrates the public health savings associated with the GHG-Step scenario alongside the "fair-share" benefits
of Federal Programs (United States Office of Management and Budget 2016) that affect California. Fair-share benefits
are calculated using the fraction of US residents living in California multiplied by the total US benefits. The GHG-
Step scenario yields benefits that are larger than those from of any program under the Federal Department of
Agriculture, Energy, Health & Human Services, Labor, and Transportation. Only the National Ambient Air Quality
Standards (NAAQS) under the US EPA have greater public health savings associated with reduced concentrations of
air pollution. As shown throughout Sect. 3, strategies to reduce GHG emissions have benefits that overlap with
NAAQS objectives and produce air quality improvements that that would otherwise be challenging or impossible to
achieve under the BAU scenario.
Taken together, the immediate and long-term savings associated with the GHG-Step scenario make a compelling case
for the shift to a low carbon energy system in California.
**4 Conclusion**
Measures to reduce GHG emissions to 80% below 1990 levels in California under the GHG-Step scenario altered
emissions of criteria pollutants (or their precursors) that generally brought nearly all regions of California into
compliance with the $O_3$ NAAQS. A few of the dense urban areas experienced minor ozone dis-benefits due to the
effects of reduced $NO_x$ concentrations leading to slightly higher ozone concentrations. Additional $O_3$ abatement
strategies may be required to offset these minor effects, but the overall improvements in $O_3$ concentrations across the
rest of the state appear to largely solve California's $O_3$ non-attainment problem. The non-linear nature of the $O_3$
response to emissions changes emphasizes the need for the research community to include realistic chemical reaction
models as a function of location in mitigation exercises.
The GHG-Step scenario reduced $PM_{2.5}$ concentrations across all regions of California through decreases in primary
emissions and secondary formation pathways. $PM_{2.5}$ concentrations increased over ocean shipping lanes in the GHG-
Step scenario but this has negligible health impact. The inland $PM_{2.5}$ reductions drive the majority of the mortality
reductions associated with the climate-friendly scenario. Total air pollution deaths in California decreased from
6,400–10,600 per year in the 2050 BAU scenario to 4,800–7,900 per year in the GHG-Step scenario. These avoided
deaths have a value of $12.2B–$20.5B per year using a Value of a Statistical Life equal to $7.6M. The avoided
mortality benefits of low carbon energy adoption in California exceed the present-day "fair share" benefits of the
combined programs under the Federal Department of Agriculture, Energy, Health & Human Services, Labor, and
Transportation. Only the National Ambient Air Quality Standards (NAAQS) under the US EPA have greater public
health benefits than adoption of low carbon energy in California. These GHG measures and air quality programs
complement and enhance one another, since adoption of Low Carbon Energy helps achieve compliance with the
NAAQS that would otherwise be challenging or impossible to achieve under the BAU scenario. The public health
benefits described here are comparable in value to published "worst-case" cost estimates for the adoption of low
carbon energy in California.  Combined with other potential long-term benefits, these immediate health benefits
strengthen the argument for the adoption of scenarios that reduce GHG emissions in California.
**5 Acknowledgements**
This study was funded by a National Center for Sustainable Transportation Dissertation Grant and the United States
Environmental Protection Agency under Grant No. R83587901. Although the research described in the article has
been funded by the United States Environmental Protection Agency it has not been subject to the Agency's required
peer and policy review and therefore does not necessarily reflect the reviews of the Agency and no official
endorsement should be inferred.

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

**Figures and Tables**

**Figure 1: Process diagram of sequence of stages for modelling and analysis.**



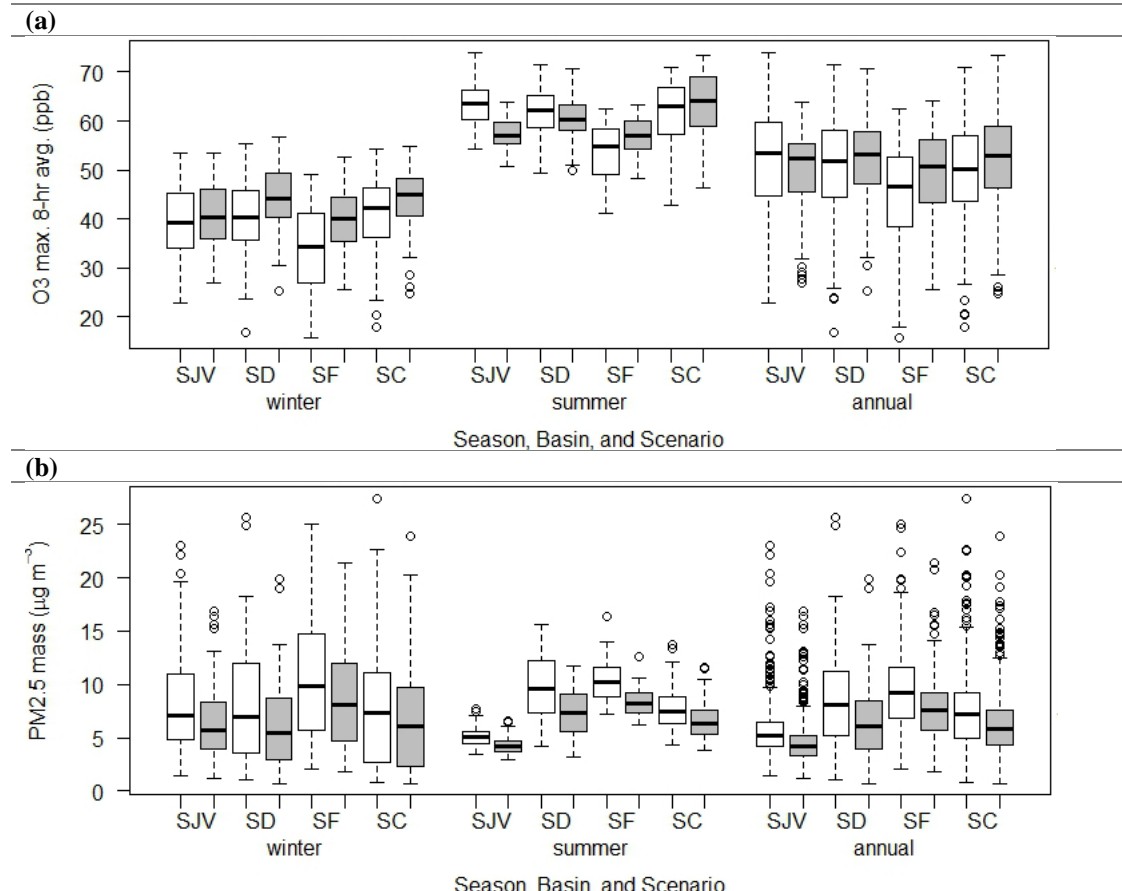

**Figure 2: (a) Population weighted 8-h average ozone concentration by region, (b) Population weighted PM$_{2.5}$ mass**
**concentration by region. Averages are shown for the winter, summer, and annual time periods in the year 2054. SJV, SD,**
**SF, SC represent the San Joaquin Valley, San Diego, San Francisco, and South Coast respectively. P-value <0.0001 was**
**found for each difference between concentrations calculated with the BAU emissions (white bars) versus the GHG-Step**
**emissions (gray bars).**

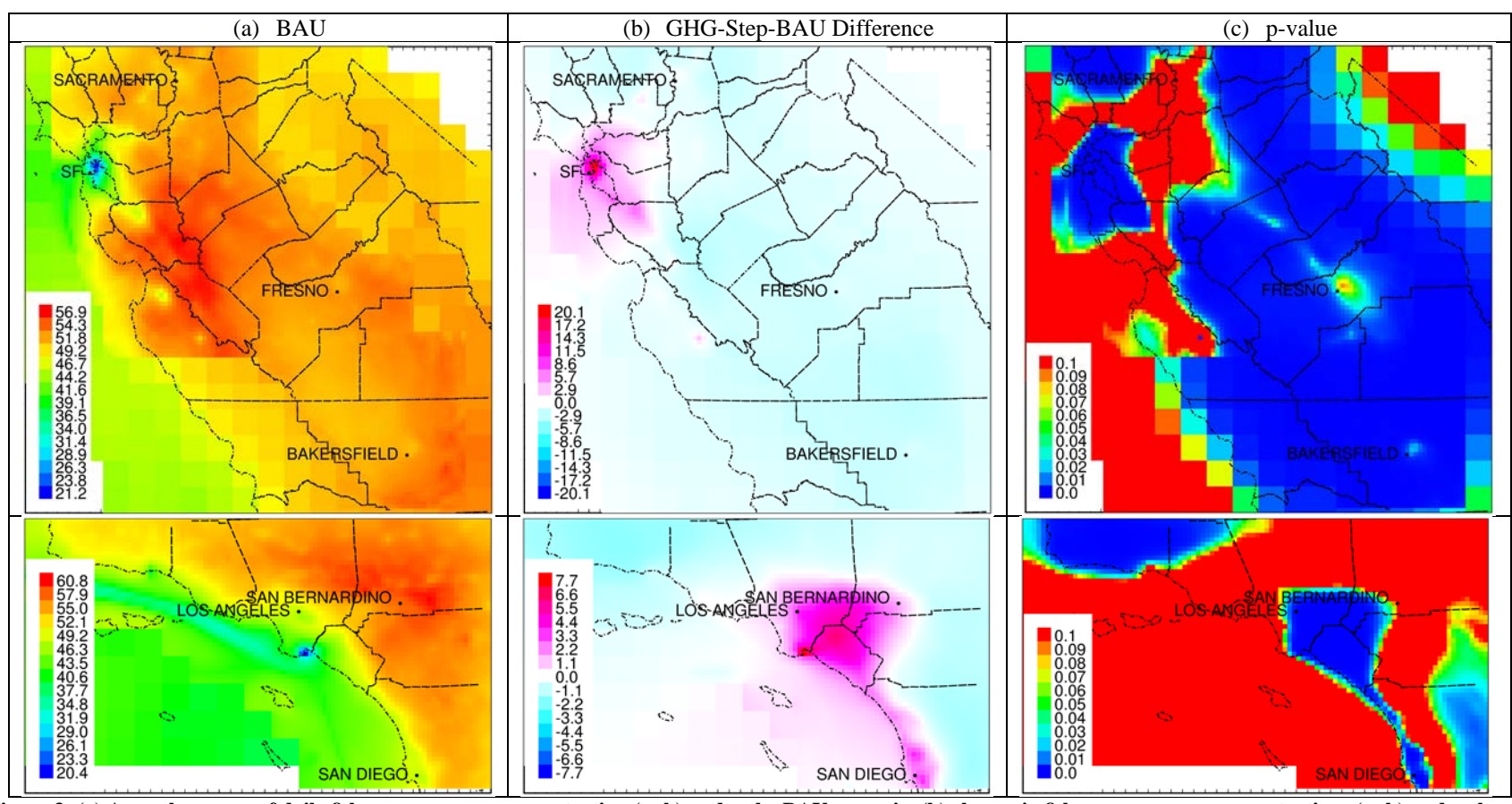

**Figure 3: (a) Annual average of daily 8-h average ozone concentration (ppb) under the BAU scenario, (b) change in 8-h average ozone concentrations (ppb) under the GHG-Step scenario, and p-value significance level of the difference between concentrations predicted using the BAU and GHG-Step scenarios. All simulations for the year 2054. Both 24 km resolution results and the finer 4 km resolution results are shown, with the finer, smaller Southern California or Central/Northern California domains are overlaid upon the coarse California domain results.**

**Table 1: the 4th highest maximum daily 8-h average ozone concentration, and number of days exceeding the standard during June-August months. Counties with 4th highest 8-h ozone concentrations ≥70 ppb are shaded in gray. See Table S4 for 2010 O$_3$ designation values and areas.**

| Basin | County or State-wide | 4th highest 8-h O$_3$ Conc. (ppb) | | | # of Days Exceeding 8-h std. of 70 ppb | | |
|---|---|---|---|---|---|---|---|
| | | 2010 | 2050 BAU | 2050 GHG-Step | 2010 | 2050 BAU | 2050 GHG-Step |
| North Central Coast (NCC) | Monterey | 75 | 72 | 64 | 12 | 3 | 0 |
| | San Benito | 97 | 75 | 65 | 44 | 31 | 0 |
| | Santa Cruz | 81 | 72 | 67 | 17 | 15 | 0 |
| South Coast (SC) | Los Angeles | 95 | 69 | 70 | 45 | 0 | 3 |
| | Orange | 92 | 63 | 70 | 43 | 0 | 4 |
| | Riverside | 123 | 80 | 79 | 62 | 47 | 43 |
| | San Bernardino | 121 | 80 | 82 | 63 | 45 | 49 |
| South Central coast (SCC) | Ventura | 83 | 66 | 63 | 46 | 0 | 0 |
| San Diego (SD) | San Diego | 93 | 68 | 67 | 48 | 1 | 2 |
| San Francisco (SF) | Alameda | 65 | 65 | 65 | 1 | 0 | 0 |
| | Contra Costa | 73 | 67 | 64 | 14 | 0 | 0 |
| | Marin | 70 | 65 | 64 | 2 | 0 | 0 |
| | Napa | 78 | 72 | 63 | 20 | 4 | 0 |
| | San Francisco | 52 | 53 | 63 | 0 | 0 | 0 |
| | San Mateo | 45 | 56 | 61 | 0 | 0 | 0 |
| | Santa Clara | 69 | 68 | 67 | 3 | 2 | 1 |
| | Solano | 82 | 71 | 64 | 36 | 10 | 0 |
| | Sonoma | 74 | 66 | 58 | 7 | 0 | 0 |
| San Joaquin Valley (SJV) | Fresno | 98 | 70 | 63 | 50 | 3 | 0 |
| | Kern | 111 | 68 | 60 | 66 | 1 | 0 |
| | Kings | 103 | 68 | 61 | 57 | 2 | 0 |
| | Merced | 98 | 71 | 63 | 59 | 5 | 0 |
| | San Joaquin | 95 | 72 | 65 | 55 | 13 | 0 |
| | Stanislaus | 100 | 71 | 65 | 63 | 7 | 0 |
| | Tulare | 112 | 71 | 62 | 70 | 6 | 0 |
| Sacramento Valley (SV) | Sacramento | 100 | 75 | 64 | 59 | 22 | 0 |
| California (CA) | State-wide | 87 | 66 | 66 | 42 | 0 | 0 |

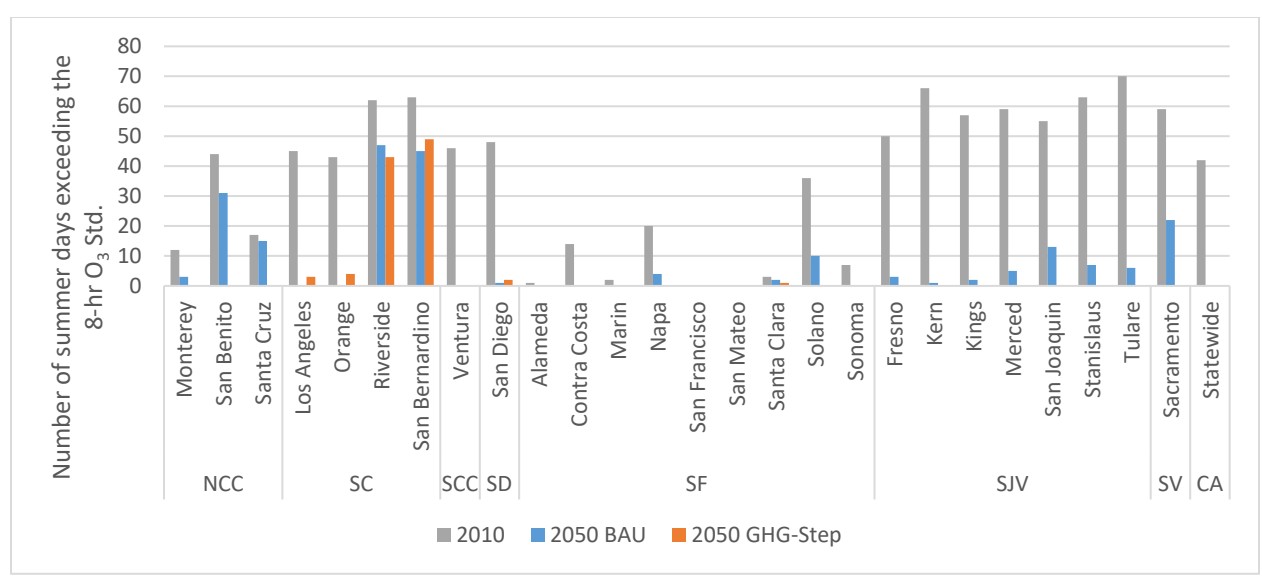

**Figure 4: Number of days in the months of June-August 2054 in which the county population-weighted daily maximum 8-h average ozone concentration exceeds the 8-h ozone NAAQS of 70 ppb for each current and future year scenario.**

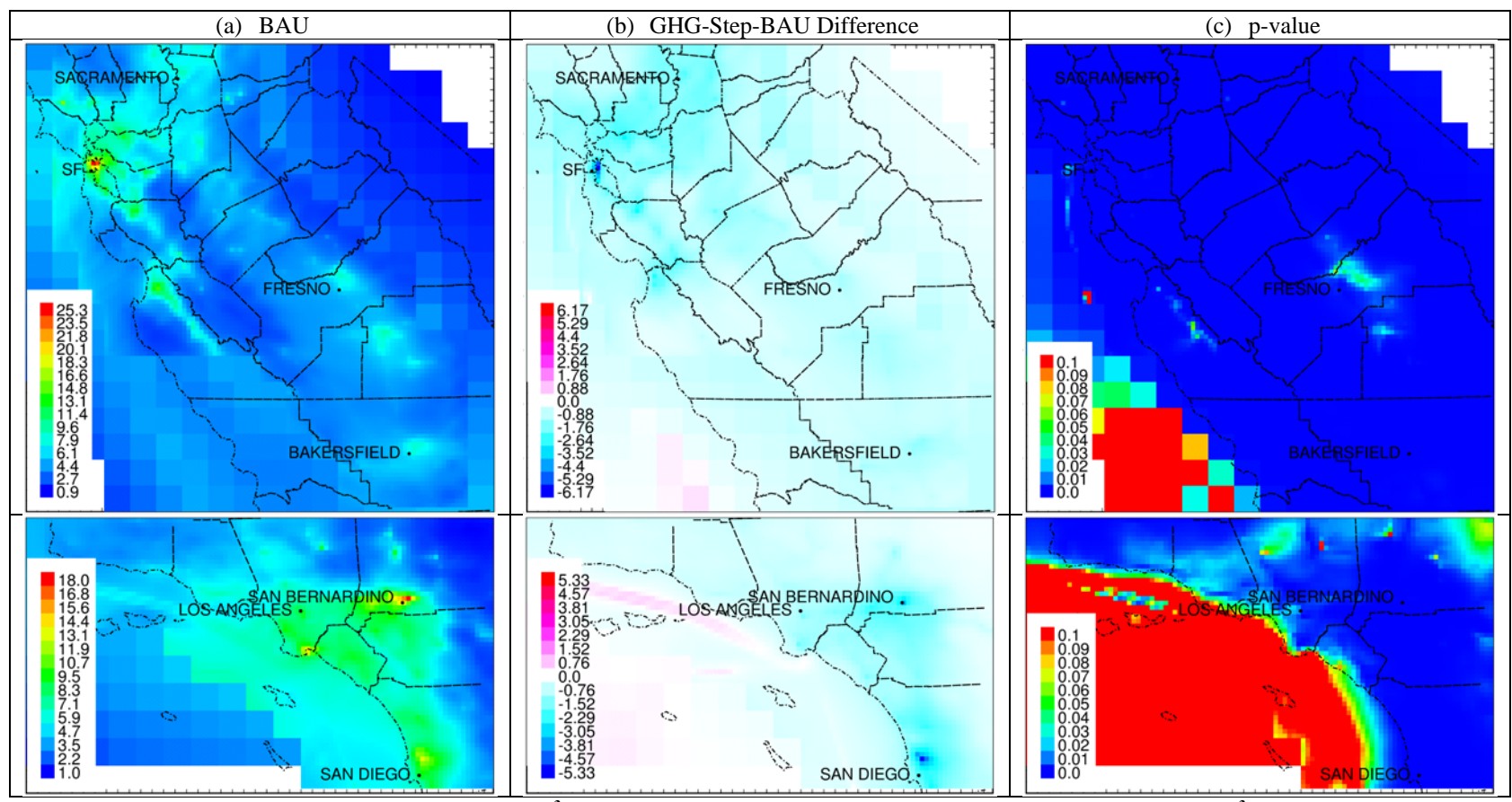

**Figure 5: (a) Annual average PM$_{2.5}$ mass concentration (µg m$^{-3}$) under the BAU scenario, (b) change in PM$_{2.5}$ mass concentrations (µg m$^{-3}$) under the GHG-Step scenario, and p-value significance level of the difference between concentrations predicted using the BAU and GHG-Step scenarios. All simulations for the year 2054. Both 24 km resolution results and the finer 4 km resolution results are shown, with the finer, smaller Southern California or Central/Northern California domains are overlaid upon the coarse California domain results.**

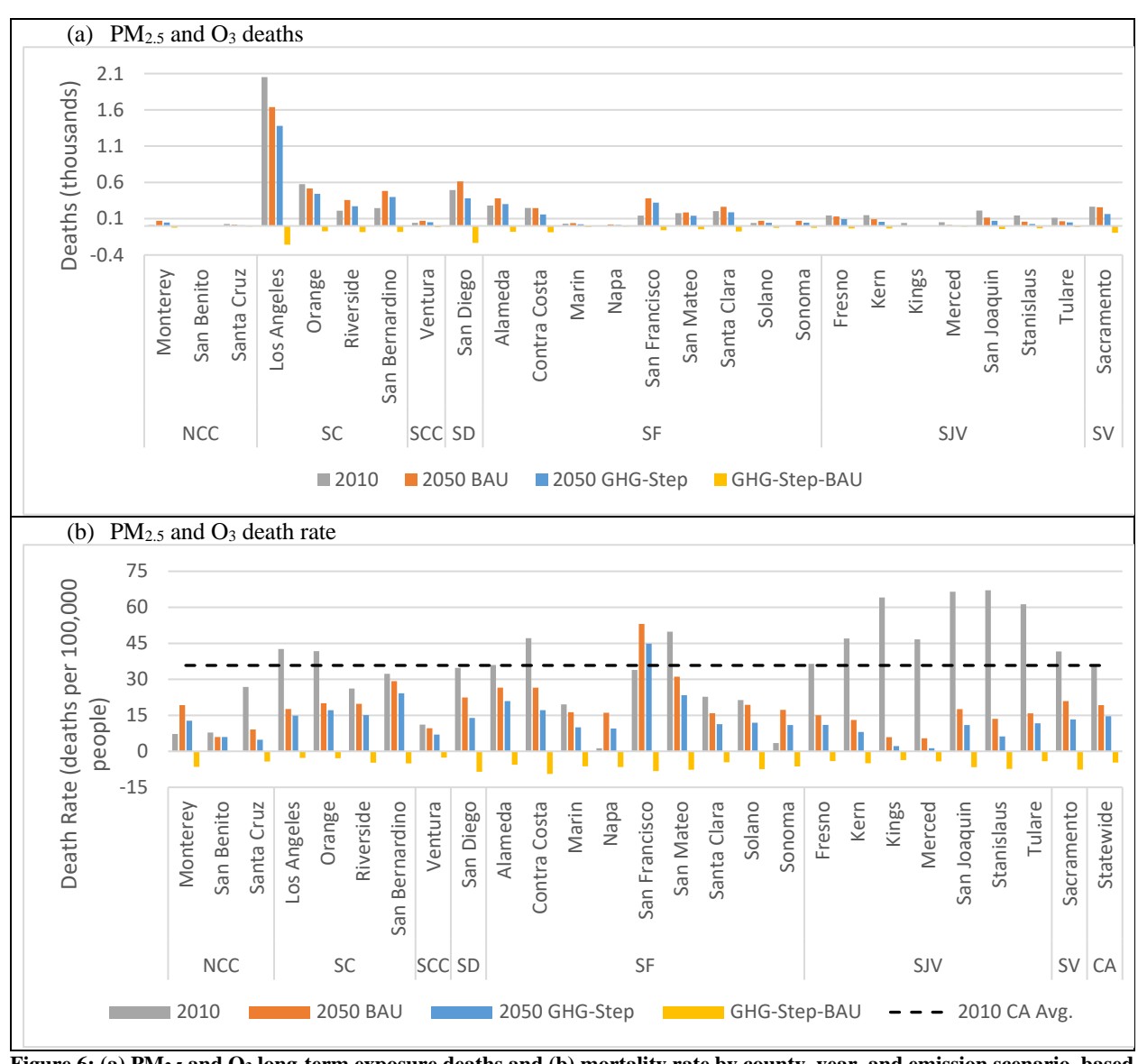

**Figure 6: (a) PM₂.₅ and O₃ long-term exposure deaths and (b) mortality rate by county, year, and emission scenario, based on combined Krewski et al. 2009 all-cause deaths associated with PM₂.₅ risk ratio (RR) and Jerrett et al. 2009 respiratory deaths associated with ozone RR.**

**(a) Deaths and cost**                    **(b) Death Rate**

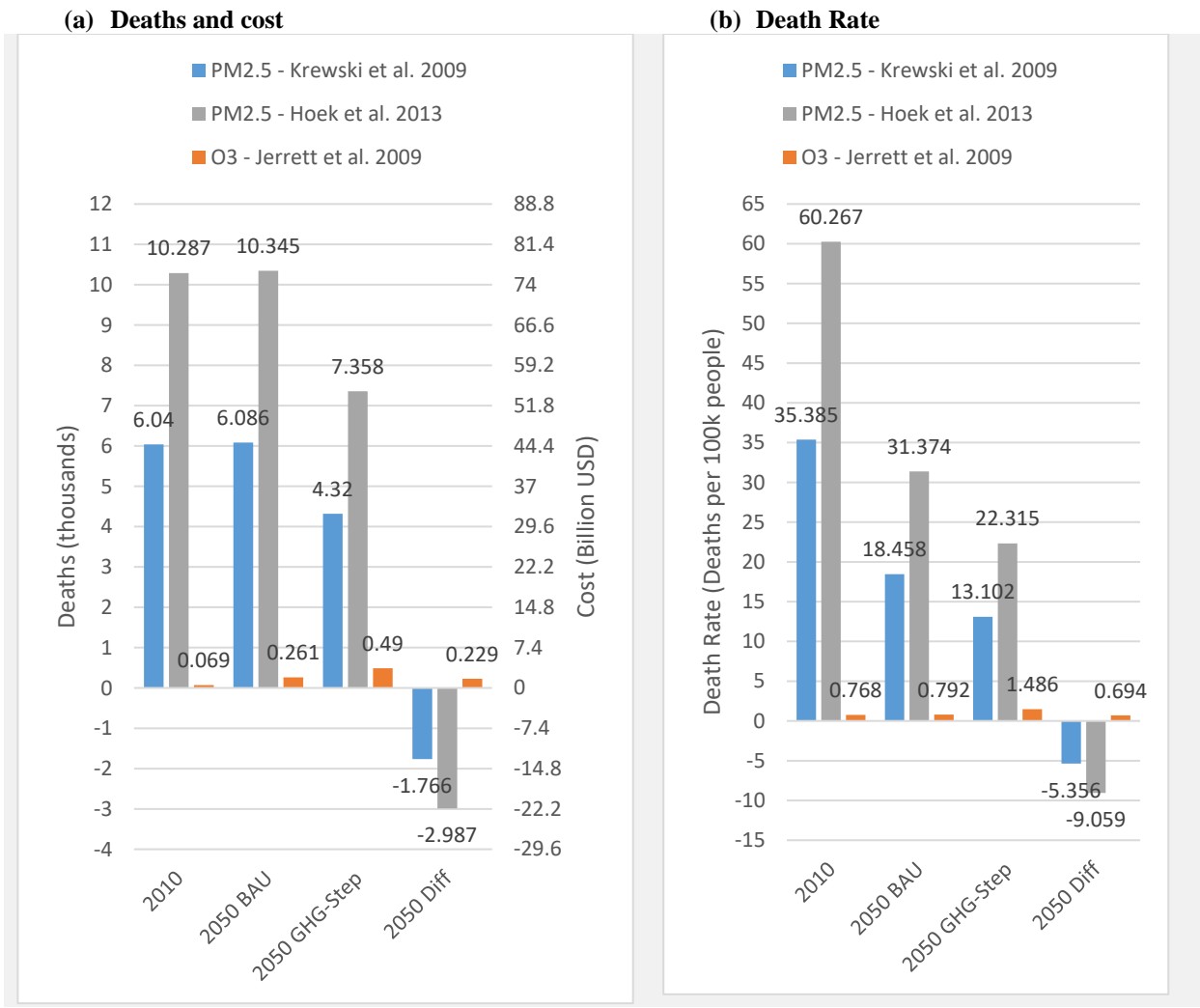

Figure 7: (a) deaths and cost and (b) death rate for the high-resolution modeling domains covering 93 % of California's population. $PM_{2.5}$ damages are estimated using methods derived by Krewski et al. 2009 (blue bars) and Hoek et al. 2013 (gray bars). Ozone damages are estimated using the methods derived by Jerrett et al. 2009 (orange bars). Only bars with the same color should be compared between 2010, 2050 BAU, and 2050 GHG-Step. The "2050 Diff" category shows the difference between the 2050 GHG-Step and BAU scenarios.

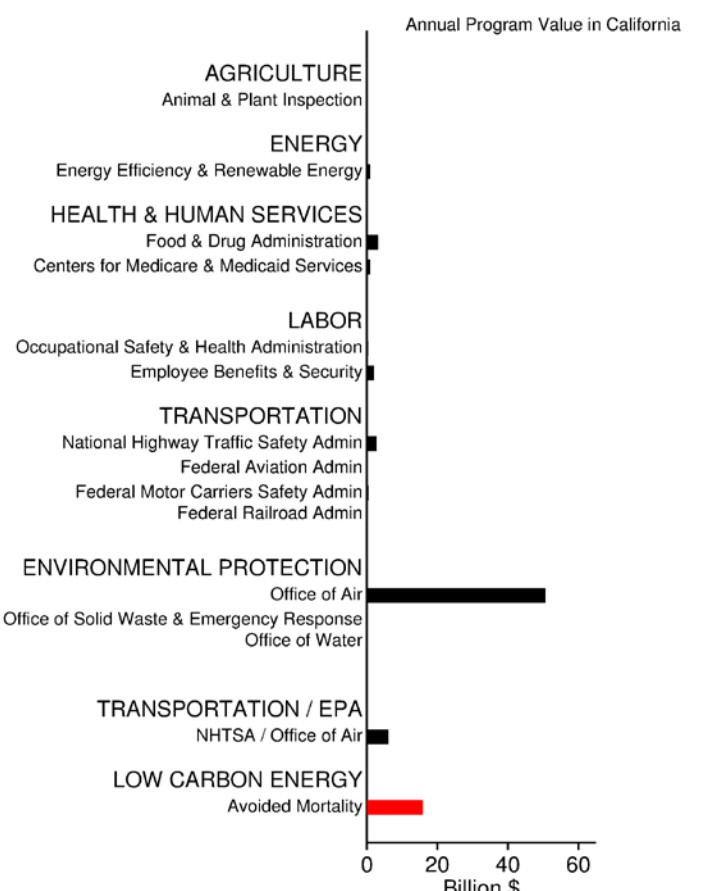

**Figure 8: Annual "fair-share" benefits of Federal programs that affect California in 2016. Fair-share fraction of US total is proportional to fraction of US population living in California. "Low Carbon Energy" represents the difference between the 2050 GHG-Step-BAU scenarios calculated in the present study.**

