# Peer review of "Low Carbon Energy Generates Public Health Savings in California"

_Atmospheric Chemistry and Physics, 2017_

## Referee Comment (RC1) · Anonymous Referee #1 · 18 Nov 2017

This paper discusses the air quality benefits of reducing greenhouse gas emissions 80% by 2050 in California. It also analyzes the cost of transition. The paper adds to the body of literature on the health benefits of transitioning from fossil fuels. Results depend highly on the scenario chosen and costs, so I will focus on these issues.

Abstract. Please state exactly what the percent reduction in emissions you are simulating between today at 2050 for some major health-affecting chemicals. The health benefits are not useful unless this information is provided side-by-side.

Health cost estimates account only for mortality. The literature suggests that morbidities and non-health damage are responsible for additional costs and that today's VSL will increase by 2050 (e.g., Jacobson et al., 2015 (cited in the MS), Section 8). The authors should account for these three factors since using VSL alone from today without

morbidity costs, non-health costs, or cost changes over time is an oversimplification. Further, low, medium, and high estimates should be used.

The authors cite a paper to claim that "recent analysis suggests that incorporating a broader range of technologies are more realistic." Aside from the fact that that paper is a criticism of another paper, so the authors, at a minimum should cite the response as well (Jacobson et al., 2017), the response contradicts the claim that a broader range is more realistic, quoting, for example, IPCC which states, "Without support from governments, investments in new nuclear power plants are currently generally not economically attractive within liberalized markets." Clearly, the use of an "economically-unattractive" resource such as nuclear is not "more realistic."

Further, California has no plans to build a new nuclear power plant or to implement coal-CCS and in fact has plans to shutter its last nuclear power plant. To the contrary, California has a proposed law (SB100) that has passed the State Senate that proposes that 60% of all electric power in the state by from approved renewable energy sources (which exclude large-scale hydro) by 2030 and the rest, by 2045, from either approved renewable energy resources or large-scale hydro (referred to in the bill as zero-carbon energy resources), not nuclear, which is being shut down in the state. The authors should clarify the situation in California, since otherwise, the results will be dated and less useful.

Along these lines, the scenarios are a bit unclear. If they are for 80% reductions in GHGs below 1990 levels by 2050, please provide a table of the emissions of relevant chemicals today and in 2050 to get a better indication of the actual reductions by chemical. Also, are you reducing only GHGs or also non-GHG air pollutants (NOx, SOx, ROGs, CO, BC, POC, other aerosol components) or are changes in GHGs affecting these pollutants only indirectly? Section 2.1 indicates that PM2.5 is being reduced only 4%, which suggests that it's emissions are not directly being controlled and that an 80% reduction in GHGs does not mean a transformation of transportation to electric vehicles or industry from combustion to electricity. How reasonable is this assumption,

particularly given CARB's intention to ratchet down emissions significantly in the next 10 years? Also, how would SB100 change electric sector emissions (thus impact your results)?

The air quality modeling appears solid.

Reference Jacobson, M.Z., M.A. Delucchi, M.A. Cameron, and B.A. Frew, The United States can keep the grid stable at low cost with 100% clean, renewable energy in all sectors despite inaccurate claims, Proc. National Acad. Sci., 114, ES021-ES023, doi:10.1073/pnas.1708069114, 2017.

---

## Referee Comment (RC2) · Anonymous Referee #2 · 21 Nov 2017

This manuscript uses an emission model to predict emissions of pollutants associated with two scenarios, including a greenhouse gas mitigation scenario, and then uses a chemical transport model to evaluate the changes in PM2.5 and ozone and its public health savings. In general, the manuscript is well organized and written. It is a good template for other related environmental health studies, so major concern is some descriptions of methodology is a bit difficult to follow. I would suggest the authors add more details.

Specific comments:

1. In the abstract, the authors mention that meteorological inputs for the year 2054 were generated under RCP 8.5 future climate, but in the manuscript there is no description about how this was done. In my understanding, GCM simulations under RCP8.5

provide boundary and initial conditions for WRF here, but different GCMs would show huge differences, which will significantly affect the results shown in this paper. I would suggest the authors add more description here about which GCM results are used and acknowledge the limitations.

2. Line 123-126 is very confusing. Please clarify why 2054 is selected.

3. Sect. 2.1: it would be better to provide spatial plots of emission changes. Is spatial allocation of emissions expected in future scenarios in the emission model?

4. Sect. 2.5: The population projection is oversimplified here. The health benefits would largely rely on changes in population. Please provide spatial map of changes in population and add more discussions on how the results are affected by population.

5. Line 178-179: The assumption of 3ug/m3 for PM2.5 and 35ppb for ozone is not solid here, which can be tuned to change the conclusions. If you use a higher threshold here, the public health benefits could be lower than cost estimates. Please provide more evidence of why these values are used.

---

## Author Comment (AC1) · 8 Feb 2018

Reviewer 1 This paper discusses the air quality benefits of reducing greenhouse gas emissions 80% by 2050 in California. It also analyzes the cost of transition. The paper adds to the body of literature on the health benefits of transitioning from fossil fuels. Results depend highly on the scenario chosen and costs, so I will focus on these issues.

Comment 1: Abstract. Please state exactly what the percent reduction in emissions you are simulating between today at 2050 for some major health-affecting chemicals. The health benefits are not useful unless this information is provided side-by-side.

Response 1: A sentence has been added to the abstract: "Net emissions reductions

across all sectors are -36% for PM0.1, -3.6% for PM2.5, mass, -10.6% for PM2.5 EC, -13.3% for PM2.5 OC, -13.7% for NOx, and -27.5% for NH3."

Comment 2: Health cost estimates account only for mortality. The literature suggests that morbidities and non-health damage are responsible for additional costs and that today's VSL will increase by 2050 (e.g., Jacobson et al., 2015 (cited in the MS), Section 8). The authors should account for these three factors since using VSL alone from today without morbidity costs, non-health costs, or cost changes over time is an oversimplification. Further, low, medium, and high estimates should be used.

Response 2: An explicit equation describing how to account for assumptions about future discount rates has been added on line 194 of the revised manuscript. We have chosen to express VSL estimates as a simple constant of $7.6M to avoid speculation about future discount rates. The appropriate choice of discount rate is subjective and best left to the reader.

"This value can be adjusted to a future year with an average discount rate "I" by multiplying with the value (1+i)future year-base year where base year is 2006. "

Low and high estimates in the mortality rates are shown in Figure 7 and summarized by the range of damage values listed in the abstract and Figure 7. Previous studies show that the medium value for damages falls near the average of the high and low estimates (within 3-15%). This point has been clarified on line 307 of the revised manuscript.

The US EPA estimates that excess deaths cause >90% of monetized damage associated with air pollution. Recommended practice from EPA is to quantify mortality risk reduction using the average Value of a Statistical Life (VSL) which is estimated based on the willingness to pay for small reductions in mortality risk associated through the selection of different job types. "Willingness to pay" estimates are thought to incorporate "cost of illness" including morbidity. Thus, the current approach accounts for mortality and morbidity which captures »90% of the monetized damage. These points have been clarified on line 195 of the revised manuscript.

https://www.epa.gov/benmap/how-benmap-ce-estimates-health-and-economic-effects-air-pollution https://www.epa.gov/environmental-economics/mortality-risk-valuation

Comment 3: The authors cite a paper to claim that "recent analysis suggests that incorporating a broader range of technologies are more realistic." Aside from the fact that that paper is a criticism of another paper, so the authors, at a minimum should cite the response as well (Jacobson et al., 2017), the response contradicts the claim that a broader range is more realistic, quoting, for example, IPCC which states, "Without support from governments, investments in new nuclear power plants are currently generally not economically attractive within liberalized markets." Clearly, the use of an "economically- unattractive" resource such as nuclear is not "more realistic."

Response 3: We agree that citing the response to the reference is an appropriate step to acknowledge the ongoing debate in the literature. We have made this addition on line 58 of the revised manuscript. We respectfully decline to enter into this debate by making further comments on either paper.

Comment 4: Further, California has no plans to build a new nuclear power plant or to implement coal-CCS and in fact has plans to shutter its last nuclear power plant. To the contrary, California has a proposed law (SB100) that has passed the State Senate that proposes that 60% of all electric power in the state by from approved renewable energy sources (which exclude large-scale hydro) by 2030 and the rest, by 2045, from either approved renewable energy resources or large-scale hydro (referred to in the bill as zero-carbon energy resources), not nuclear, which is being shut down in the state. The authors should clarify the situation in California, since otherwise, the results will be dated and less useful.

Response 4: We have added the following statement on line 119 of the revised manuscript "The future BAU and GHG-Step scenarios considered in the present study do not include nuclear or coal-fired (with or without CCS) electricity generation in California."

Comment 4a: Along these lines, the scenarios are a bit unclear. If they are for 80% reductions in GHGs below 1990 levels by 2050, please provide a table of the emissions of relevant chemicals today and in 2050 to get a better indication of the actual reductions by chemical.

Response 4a: Emissions reductions are now summarized in the abstract and on line 113 of the revised manuscript as requested.

Comment 4b: Also, are you reducing only GHGs or also non-GHG air pollutants (NOx, SOx, ROGs, CO, BC, POC, other aerosol components) or are changes in GHGs affecting these pollutants only indirectly?

Response 4b: We focus on changes to criteria pollutants (NOx, SOX, ROGs, PM components) as summarized in the abstract and on line 113 of the revised manuscript.

Comment 4c: Section 2.1 indicates that PM2.5 is being reduced only 4%, which suggests that it's emissions are not directly being controlled and that an 80% reduction in GHGs does not mean a transformation of transportation to electric vehicles or industry from combustion to electricity. How reasonable is this assumption, particularly given CARB's intention to ratchet down emissions significantly in the next 10 years?

Response 4c: The scenarios considered in this paper have been described in great detail by a companion article in GMD referenced in section 2.1 of the current paper. The economic optimization model behind the scenarios is CA-TIMES. Proposed changes to the energy system are dramatic in all sectors. The scenarios adopt alternative fuels and great care is exercised to estimate realistic changes to criteria pollutant emissions as a result. The current paper shows that the resulting changes to PM2.5 and ozone concentrations are significant. We believe these scenarios are realistic options that serve as useful analysis points for possible changes to future air quality.

Comment 4d: Also, how would SB100 change electric sector emissions (thus impact

your results)?

Response 4d: The companion paper referenced in section 2.1 explains that electricity generation in the 2050 GHG-Step scenario is dominated by wind (34%), solar (34%), and natural gas (18%) with smaller contributions from tidal, geothermal, and hydro. These points are repeated on line 121 of the revised manuscript.

If SB100 is adopted as written, the natural gas component of this generation would shift to biogas and / or one of the other renewable sources. We believe this would not change the conclusions of the study, but we will explicitly model this scenario in future work.

Comment 5: The air quality modeling appears solid.

Response 5: Thank you.

Reference Jacobson, M.Z., M.A. Delucchi, M.A. Cameron, and B.A. Frew, The United States can keep the grid stable at low cost with 100% clean, renewable energy in all sectors despite inaccurate claims, Proc. National Acad. Sci., 114, ES021-ES023, doi:10.1073/pnas.1708069114, 2017.  

Reviewer 2

This manuscript uses an emission model to predict emissions of pollutants associated with two scenarios, including a greenhouse gas mitigation scenario, and then uses a chemical transport model to evaluate the changes in PM2.5 and ozone and its public health savings. In general, the manuscript is well organized and written. It is a good template for other related environmental health studies, so major concern is some descriptions of methodology is a bit difficult to follow. I would suggest the authors add more details.

Specific comments:

1. In the abstract, the authors mention that meteorological inputs for the year 2054

were generated under RCP 8.5 future climate, but in the manuscript there is no description about how this was done. In my understanding, GCM simulations under RCP8.5 provide boundary and initial conditions for WRF here, but different GCMs would show huge differences, which will significantly affect the results shown in this paper. I would suggest the authors add more description here about which GCM results are used and acknowledge the limitations.

Response: The text on line 131 of the revised manuscript has been expanded to clarify that: "The focus of the current study is to evaluate how the emissions changes lead to different air quality outcomes. Both emissions scenarios are evaluated using the same meteorology, which minimizes the variability introduced by the climate signal."

2. Line 123-126 is very confusing. Please clarify why 2054 is selected.

Response: The text on line 129 of the revised manuscript has been modified to state the selection criteria more clearly. "The 2054 calendar year was the median year over the period 2048–2054 for domain-average PM2.5 concentrations within the South Coast Air Basin that contains the majority of the population in California. "

3. Sect. 2.1: it would be better to provide spatial plots of emission changes. Is spatial allocation of emissions expected in future scenarios in the emission model?

Response: Spatial plots of emissions changes have been shown in a companion paper by Zapata et al (2017). This point has been clarified on line 123 of the revised manuscript.

4. Sect. 2.5: The population projection is oversimplified here. The health benefits would largely rely on changes in population. Please provide spatial map of changes in population and add more discussions on how the results are affected by population.

Response: Population affects both the emissions location and the air pollution exposure. The key requirement is to make consistent assumptions in both areas in order to get realistic population exposure. The text on line 161 of the revised manuscript has

been added to clarify this point. "Population acts as a spatial surrogate for distributing emissions and as a receptor for calculating the public health effects of air pollution. Consistent population fields were used for both of these tasks in the current study. Population growth rates by county are summarized by Zapata et al. (2017)"

5. Line 178-179: The assumption of 3ug/m3 for PM2.5 and 35ppb for ozone is not solid here, which can be tuned to change the conclusions. If you use a higher threshold here, the public health benefits could be lower than cost estimates. Please provide more evidence of why these values are used.

Response: The majority of the health damages are associated with PM2.5 and so we focus on the appropriate background concentration for this parameter. Ostro et al. performed a survey of published studies and concluded that there is no apparent lower limit for PM2.5 concentrations at which there is no health risk. Thus, the appropriate background concentration is the true PM2.5 concentration attributable to natural background processes or upwind sources. Background concentrations on the west coast of North America are often measured at mountain sites that sample the free troposphere. Herner et al. (2005) measured PM1.8 concentrations of 4 $\mu$g m-3 at Sequoia National Park (elevation 535m) during periods when this site was in the free troposphere. McKendry (2006) performed a survey of published literature and reviewed monitoring data in British Columbia on the west coast of North America and estimated that background PM2.5 concentrations are 2 $\mu$g m-3 with little evidence of change over time. Thus, the selected background value of 3 $\mu$g m-3 seems reasonable. These points have been clarified on line 187 of the revised manuscript.